# Mapping Leaf Area Index at Various Rice Growth Stages in Southern India Using Airborne Hyperspectral Remote Sensing

Mathyam Prabhakar *, Kodigal A. Gopinath, Nakka Ravi Kumar, Merugu Thirupathi, Uppu Sai Sravan, Golla Srasvan Kumar, Gutti Samba Siva, Pebbeti Chandana and Vinod Kumar Singh

ICAR-Central Research Institute for Dryland Agriculture, Santoshnagar, Hyderabad 500059, India
* Correspondence: prab249@gmail.com

**Abstract:** Globally, rice is one of the most important staple food crops. The most significant metric for evaluating the rice growth and productivity is the Leaf Area Index (LAI), which can be effectively monitored using remote sensing data. Hyperspectral remote sensing provides contiguous bands at narrow wavelengths for mapping LAI at various rice phenological stages, and it is functionally related to canopy spectral reflectance. Hyperspectral signatures for different phases of rice crop growth was recorded using Airborne Visible Near-Infrared Imaging Spectrometer-Next Generation (AVIRIS-NG) along with corresponding ground based observations. Ground-based hyperspectral canopy spectral reflectance measurements were recorded with FieldSpec 3 Hi-Res spectroradiometer (ASD Inc., Forsyth County, GA, USA; spectral range: 350–2500 nm) and LAI data from 132 farmer's fields in Southern India. Among 29 hyperspectral vegetation indices tested, 8 were found promising for mapping rice LAI at various phenological stages. Among all the growth stages, the elongation stage was the most accurately estimated using vegetation indices that exhibited a significant correlation with the airborne hyperspectral reflectance. The validation of hyperspectral vegetation indices revealed that the best fit model for estimating rice LAI was $mND_{705}$ (red-edge, blue, and NIR bands) at seedling and elongation, SAVI (red and NIR bands) at tillering and WDRVI (red and NIR bands) at booting stage.

**Keywords:** hyperspectral remote sensing; vegetation indices; canopy reflectance; leaf area

## 1. Introduction

Global rice production was 513.68 million tonnes, of which India contributes 132 million tonnes after China, and plays an important role in the global rice economy [1]. India ranks second in consumption, which attributes to the adequate supply of rice, leading to food security. Remote sensing technology has gained attention, as it provides rapid and macro-scale observations using spectral data of the crop canopy to ameliorate the crop growth parameter estimation [2–4]. The reflectance curve in the red and blue region displays a valley when the plant canopy is healthy, as it significantly absorbs blue and red light and reflects green light [5]. Thus, higher reflectance in the near infrared region (NIR) was closely associated with internal cell structure, biomass, vegetation cover, leaf water content, and LAI, whereas red-edge region exhibits strong absorption because of leaf chlorophyll content, and reflection is due to mesophyll cells [6].

The LAI of crop is a key variable for estimating the foliage cover and photosynthetic processes, and ultimately aids in forecasting of the crop growth and yield [7]. The estimation of LAI includes direct and indirect measurement methods. The direct measurement of LAI includes destructive sampling and laboratory evaluation, which is time-consuming, labor-intensive, discontinuous, and is not applicable for large-scale investigations [8,9]. Therefore, numerous research efforts have been emphasized on the indirect measurement of LAI using remote sensing technology due to its non-destructive and rapid monitoring potential [10,11]. Recently, the use of remote sensing technology to estimate LAI at various

rice growth stages has been widely developed, especially for large-scale and long-term crop monitoring [12–17].

Among various vegetation indices (VI) and neural network approaches, the scaled normalized difference vegetation index (NDVI) methodology was the most efficient way to retrieve LAI [18]. However, it is sensitive to atmospheric processes, soil conditions, and saturates at low LAI [19]. Thus, other VIs *viz.*, modified red-edge normalized difference vegetation index (mND$_{705}$) [20], soil adjusted vegetation index (SAVI) [21], triangular vegetation index [22], and renormalized difference vegetation index (RDVI) [23] were developed to enhance the precision of LAI estimation. The VIs with red-edge bands are better predictors of LAI [24–27]. Recently, hyperspectral remote sensing has drawn the emphasis on LAI estimation because it provides continuous spectral coverage and achieves a spectral resolution of <10 nm in the range of 400–2500 nm [4,28]. Hyperspectral remote sensing data is used for the estimation of rice LAI at various growth stages [29–32]. Further, hyperspectral vegetation indices were used for precise and efficient nitrogen management by estimating leaf nitrogen content in rice and wheat at various phenological stages [33–35]. Other than LAI and leaf nitrogen content, the above-ground biomass and yield estimation can also be better predicted using hyperspectral remotely sensed images [36,37]. Mapping the distribution of LAI in winter wheat for various growth phases was performed using UAV-based hyperspectral data [9] and AVIRIS-NG data [38]. Many studies in rice revealed that the VIs were positively correlated with biomass during pre-heading stages and negatively during post-heading stages [39,40].

To maximize the effectiveness of LAI estimation, identification of significant wave bands in the hyperspectral data is essential for retrieving more specific VIs [41]. Further, the combination of these wave bands under diverse environmental conditions and cultural practices provides appropriate information for the characterization of LAI with the crop phenological stages [42]. In this study, we examined the utility of different VIs through hyperspectral measurements from airborne AVIRIS-NG to map LAI at various phenological stages of rice.

## 2. Materials and Methods

### 2.1. Study Area

The study area is Banaganapalle mandal, Kurnool district, Andhra Pradesh, India, in which rice is predominantly cultivated in both wet and dry seasons. The wet season plantings were carried out during July–August, and dry season plantings during November–December. Rice was mostly transplanted, and in few cases direct seeding was noticed in the study area. The soils are predominantly vertisols, and rice was grown following the recommended agronomic practices. Farmers mostly cultivate rice varieties *viz.*, BPT 5204, NDLR-7, RP Bio-226, and BPT 2270 of medium to long (130–160 days) crop growth duration.

### 2.2. Sampling Sites and Field Data Collection

Ground-truth surveying of rice fields was carried out during the fourth week of February 2018 (20–24 February), and the AVIRIS-NG images were acquired on 26 February 2018, which has spectral range of 380–2510 nm with spatial resolution of 4–8 m. The AVIRIS-NG map with six scenes used in the study is depicted in Figure 1. Farmers' fields were surveyed based on the stratified random sampling procedure. A total of 132 fields in Banaganapalle were selected randomly during the dry season for rice LAI estimation at various phenological phases (15.723°N to 15.979°N latitude; 77.926°E to 78.186°E longitude) (Figure A1 in Appendix A, Figure 2). The dates of sowing and transplanting were recorded through farmer's interaction and rice growth stages observed at the time of data collection. The plantings varied considerably, and lasted for three months from the second week of November 2017 to first week of February 2018. The harvesting was done for a period of two months, i.e., from the first week of March to the second week of May 2018.

Field measurements of in situ LAI was recorded from the surveyed fields individually during or near the AVIRIS-NG airborne data acquisition.

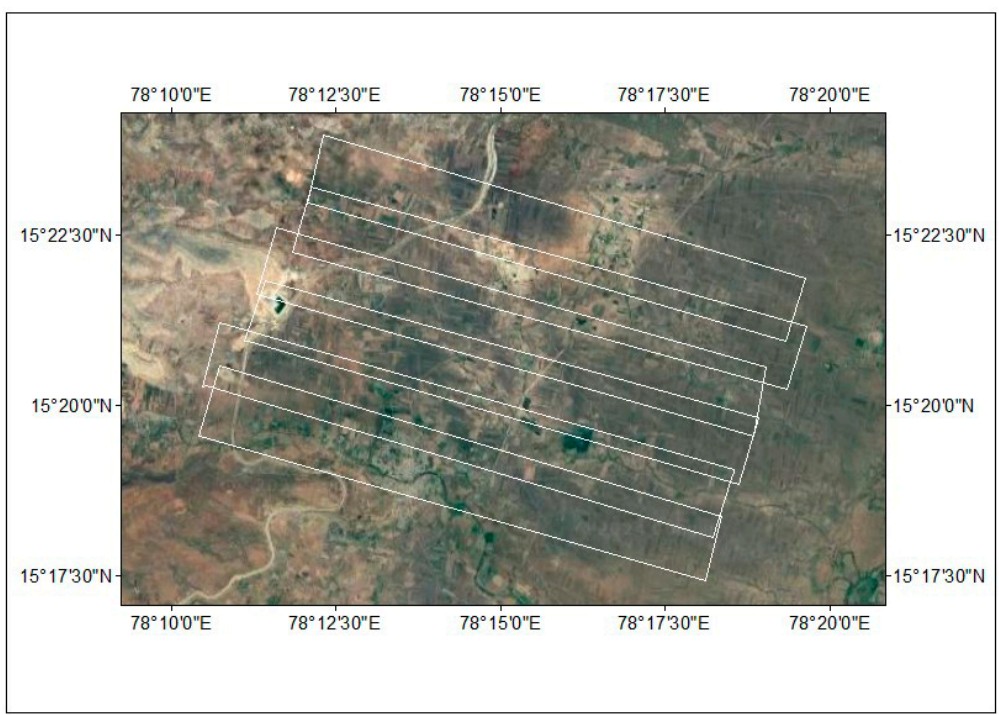

**Figure 1.** AVIRIS-NG map with six scenes used for the study (Digitization footprint: 1.85 MB/ha).

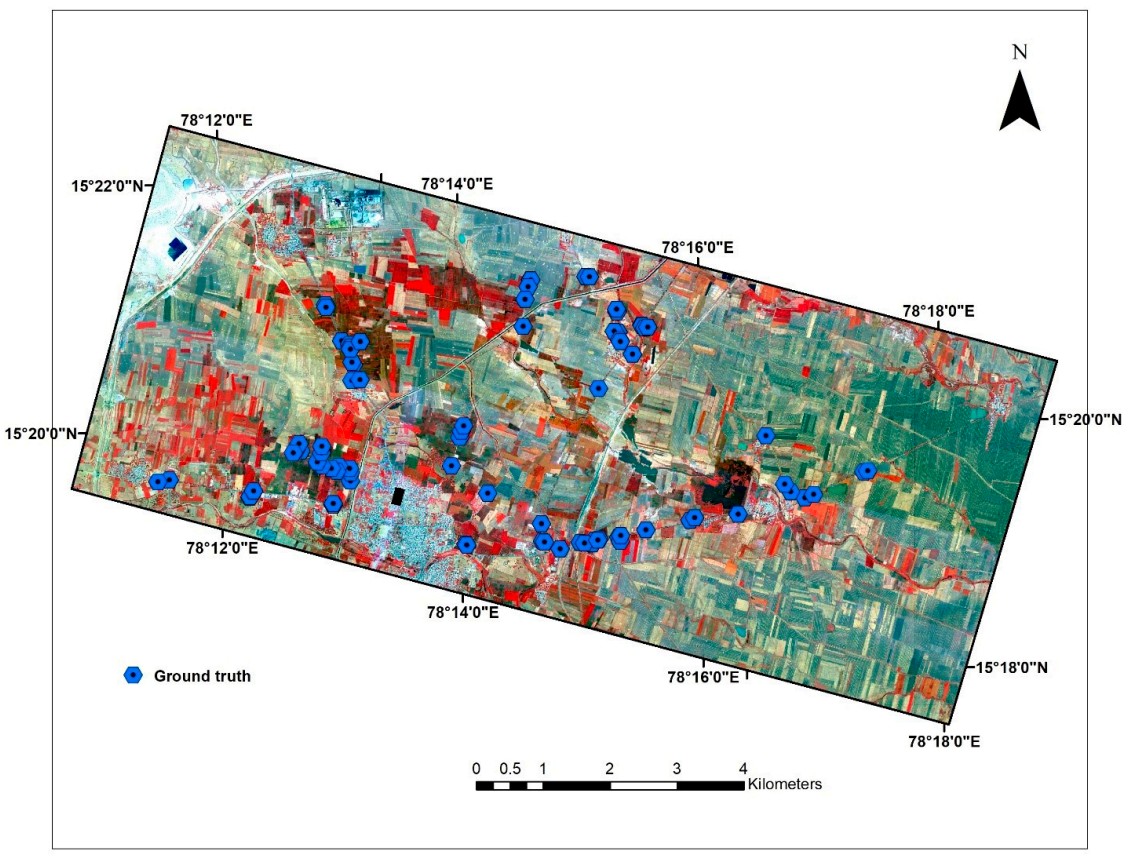

**Figure 2.** FCC of the study area and the sampling sites with different color codes (Red: 817 nm, Green: 642 nm, and Blue: 432 nm).

### 2.3. Sampling of LAI

Field surveys were conducted from 132 farmer's fields for ground-truth data collection. As the study area was a traditional rice growing belt, the crop was observed at different phenological stages. Accordingly, the fields were categorized into seven phenological stages (Table 1) as seedling (planted in the first to second week of February), tillering (planted in the fourth week of January), elongation (planted during the second week of January), booting (planted in the first week of January), heading (planted during the fourth week of December), flowering (planted during the second week of December), and maturity (planted during the second week of November) (Figure 3). The surveyed fields were segregated separately for calibration (78 fields) and validation (54 fields) to estimate LAI using remote sensing (Table 1). Five plants were selected randomly from each surveyed field for non-destructive LAI measurement using SunScan plant canopy analyzer (Delta-T devices Ltd., Burwell, UK) representing each phenological stage.

**Table 1.** Details of rice fields sampled at different phenological stages.

| Crop Growth Stage | Field Samples Considered for Calibration | Field Samples Considered for Validation |
| --- | --- | --- |
| Seedling | 14 | 12 |
| Tillering | 19 | 12 |
| Elongation | 9 | 7 |
| Booting | 11 | 7 |
| Heading | 10 | 6 |
| Flowering | 8 | 5 |
| Maturity | 7 | 5 |
| Total | 78 | 54 |

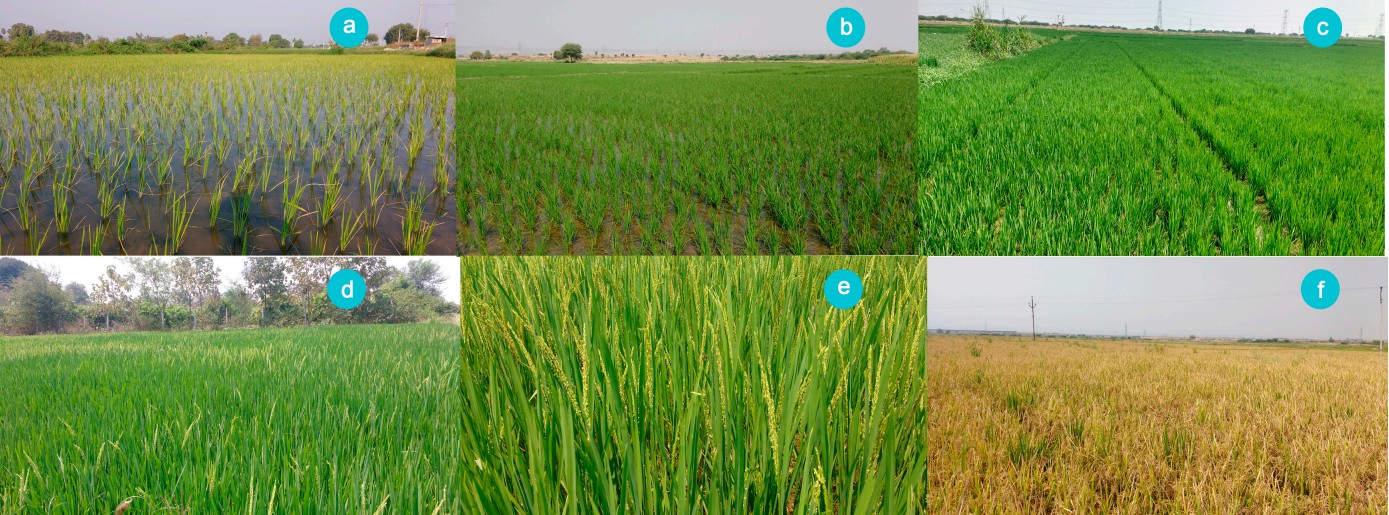

**Figure 3.** Rice crop at various growth phases: (**a**) seedling; (**b**) tillering; (**c**) elongation; (**d**) heading; (**e**) flowering; (**f**) maturity.

### 2.4. Canopy Spectral Reflectance Measurements

Canopy spectral reflectance measurements from rice were recorded with FieldSpec 3 Hi-Res spectroradiometer (ASD Inc., Falls Church, VA, USA; spectral range: 350–2500 nm). All the spectral measurements were made during 20–24 February 2018 between 11:00 and 14:00 h Indian standard time (GMT + 5.30 h) under clear sky. The radiometer was configured to record 30 scans for each sample, and 15–25 plants from each field were sampled at the time of the data recording. The pistol grip of radiometer was positioned at 1 m above the crop canopy, centered over the rice hills, with 25° FOV fore optic to ensure the reflectance from a single desired plant. Prior to field measurements, the spectral radiometer was warmed-up for about 20 min to

avoid the spectral steps at detector overlap wavelength regions, which occur due to different warm-up rates for the three spectroradiometer arrays [43]. The sampling interval was 1.4 nm at 350–1000 nm range and 2 nm at 1000–2500 nm range. The spectral resolution (full-width half-maximum) was 3 nm at 700 nm, 6.5 nm at 1400 nm, and 8.5 nm at 2100 nm. Finally, the collected spectral data was exported to RS3 software 5.6.4 and interpolated using ASD ViewSpecPro software 5.6.10 in the post processing to produce values at each nanometer [44]. The spectral reflectance data from 1300 to 1400 nm and 1800 to 1950 nm were not considered for analysis due to noise caused by atmospheric water absorption in the raw spectrum [45,46].

### 2.5. AVIRIS-NG Airborne Data Acquisition

The present study is part of the collaborative program between Indian Space Research Organization (ISRO), India and Jet Propulsion Laboratory (JPL), NASA; Department of Science & Technology, Government of India, and many other research organizations including the Indian Council of Agricultural Research (ICAR). AVIRIS-NG data covers a wavelength range of 380–2510 nm in VNIR and SWIR spectral channels, with a contiguous spectral band of 425 at a 5 nm interval. Spatially, it has a varying resolution of 4–8 m with a flying altitude of 4–8 km for a swath of 4–6 km. The AVIRIS-NG radiance data were orthorectified using input geometry files (IGM) present with the data containing the original line number and sample number for georeferencing, and later atmospheric correction was carried out [47,48].

### 2.6. Data Processing

AVIRIS-NG data Endmember spectra was collected using ENVI software 4.7 for the selected locations in farmers' fields. Correlation analysis was performed for spectral reflectance and LAI. Highly correlated spectral bands were used to calculate the new band ratios and estimate the relevant vegetation indices. The regression analysis was performed between vegetation indices and LAI. Descriptions and formulas of VIs [20,35,49] used in the study are listed in Table 2.

**Table 2.** Hyperspectral vegetation indices used in the study.

| Spectral Vegetation Index | Formula |
|---|---|
| Vogelmann Index (VOG 1) | $(R_{740}/R_{720})$ |
| Meris Terrestrial Chlorophyll Index (MTCI) | $(R_{850} - R_{730})/(R_{730} - R_{675})$ |
| Vogelmann Index (VOG 2) | $(R_{734} - R_{747})/(R_{715} + R_{726})$ |
| Modified simple ratio (MSR) | $\dfrac{((R_{800}/R_{670}) - 1)}{\left(\sqrt{(R_{800}/R_{670})} + 1\right)}$ |
| Modified Red-edge Normalized Difference Vegetation Index (mND$_{705}$) | $\dfrac{(R_{750} - R_{705})}{(R_{750} + R_{705} - 2*R_{445})}$ |
| Double Difference Index (DD) | $(R_{749} - R_{720}) - (R_{701} - R_{672})$ |
| Green Normalized Difference Vegetation Index (GNDVI) | $(R_{750} - R_{550})/(R_{750} + R_{550})$ |
| Soil Adjusted Vegetation Index (OSAVI) | $(1 + 0.16)(R_{800} - R_{670})/(R_{800} + R_{670} + 0.16)$ |
| Renormalized Difference Vegetation Index (RDVI) | $\dfrac{R_{800} - R_{670}}{\sqrt{R_{800} + R_{670}}}$ |
| Simple Ratio (SR) | $R_{800}/R_{670}$ |
| Modified Triangular Vegetation Index (MTVI 2) | $\dfrac{1.5*1.2*(R_{800} - R_{550}) - 2.5*(R_{670} - R_{550})}{\sqrt{(2*R_{800} + 1)^2 - (6*R_{800} - 5*\sqrt{R_{670}}) - 0.5}}$ |
| Soil-Adjusted Vegetation Index (SAVI) | $\left(\dfrac{R_{800} - R_{670}}{R_{800} + R_{670} + 0.5}\right)(1 + 0.5)$ |
| Normalized Difference Vegetation Index (NDVI) | $(R_{810} - R_{680})/(R_{810} + R_{680})$ |
| Ratio Vegetation Index (RVI) | $(R_{810}/R_{680})$ |
| Enhanced Vegetation Index (EVI 1) | $2.5*(R_{860} - R_{645})/(1 + R_{860} + 6*R_{645} - 7.5*R_{470})$ |

**Table 2.** *Cont.*

| Spectral Vegetation Index | Formula |
|---|---|
| Difference Vegetation Index (DVI) | $(R_{810} - R_{680})$ |
| Photochemical Refectance Index (PRI) | $(R_{570} - R_{531}) / (R_{570} + R_{531})$ |
| Transformed Vegetation Index (TVI) | $\dfrac{120 * (R_{750} - R_{550}) - 200 * (R_{670} - R_{550})}{2}$ |
| New Double Difference Index (DDn) | $2 * R_{710} - R_{660} - R_{760}$ |
| Modified Red-edge Simple Ratio Index (MSR$_{705}$) | $(R_{750} - R_{445}) / (R_{705} - R_{445})$ |
| Modified Non-Linear Index (MNLI) | $\dfrac{(1 + 0.5) * (R_{810}^2 - R_{680})}{(R_{810}^2 + R_{680} + 0.5)}$ |
| Structure Insensitive Pigment Index (SIPI) | $(R_{800} - R_{445}) / (R_{800} - R_{680})$ |
| Water Index (WI) | $R_{900} / R_{970}$ |
| Red-Edge Vegetation Stress Index (RVSI) | $\dfrac{R_{714} - R_{752}}{2} - R_{733}$ |
| Standardized LAI Determining Index (SLAIDI) | $S * \dfrac{R_{1050} - R_{1250}}{R_{1050} + R_{1250}} * R_{1555}$ where $S = 5$ |
| Normalized Difference Water Index (NDWI) | $(R_{858} - R_{2130}) / (R_{858} + R_{2130})$ |
| Normalized Difference Infrared Index (NDII) | $(R_{819} - R_{1600}) / (R_{819} + R_{1600})$ |
| Red Edge Position Index (REP) | $R_{700} + 40 * \left[ \dfrac{(R_{670} + R_{780})}{2} - R_{700} \right] / (R_{740} - R_{700})$ |
| Wide Dynamic Range Vegetation Index (WDRVI) | $\dfrac{(0.2 * R_{800} - R_{670})}{(0.2 * R_{800} + R_{670})} + \dfrac{(1 - 0.2)}{(1 + 0.2)}$ |

*2.7. Data Analysis*

The mean reflectance values of LAI at different phenological stages was obtained from different rice fields (area ranged from 0.5 to 10.0 ha) to find out the sensitive wavebands for each phenological stage. The descriptive statistics of LAI and vegetation indices at different phenological stages were performed, and means along with the standard deviation were presented. The spectral angle mapper (SAM) matching algorithm was used for comparing the similarity between ground-truth and AVIRIS-NG image spectra [50]. Coefficient of determination ($R^2$) and root mean square error (RMSE) were used to evaluate the prediction accuracy of the regression models [51], as they are the indicators of how well the regression models (best-fit function) capture the relationship between LAI and vegetation indices. Regression analysis was performed between LAI and vegetation indices for calibration and validation of the data, and $R^2$ and RMSE were presented. Statistical analysis was performed using SAS Institute Inc., (Cary, NC, USA) [52].

**3. Results**

*3.1. Reflectance Spectra of Rice Canopy from Hand Held Hyperspectral Radiometry*

The ground-based mean canopy reflectance spectra of rice at different phenological stages was recorded and presented in Figure 4a. It was found that the spectral reflectance of rice showed variation at various phenological stages in ultraviolet (350–400 nm), visible (400–750 nm), and NIR regions (750–1100 nm) with maximum reflectance in the NIR region. In UV region, the canopy reflectance was maximum at tillering followed by seedling stage and the lowest was observed during flowering and maturity. Within the visible region, green region (490–560 nm) had higher reflectance compared to violet blue (400–425 nm) and red region (640–685 nm). The reflectance in visible region increased as the crop advanced from seedling to tillering and reduced from elongation to flowering, with the lowest reflectance at heading. The reflectance in the NIR region varied steadily with crop phenological stage, indicating a distinct variation in the spectral pattern that can be leveraged to identify the crop phenological stage.

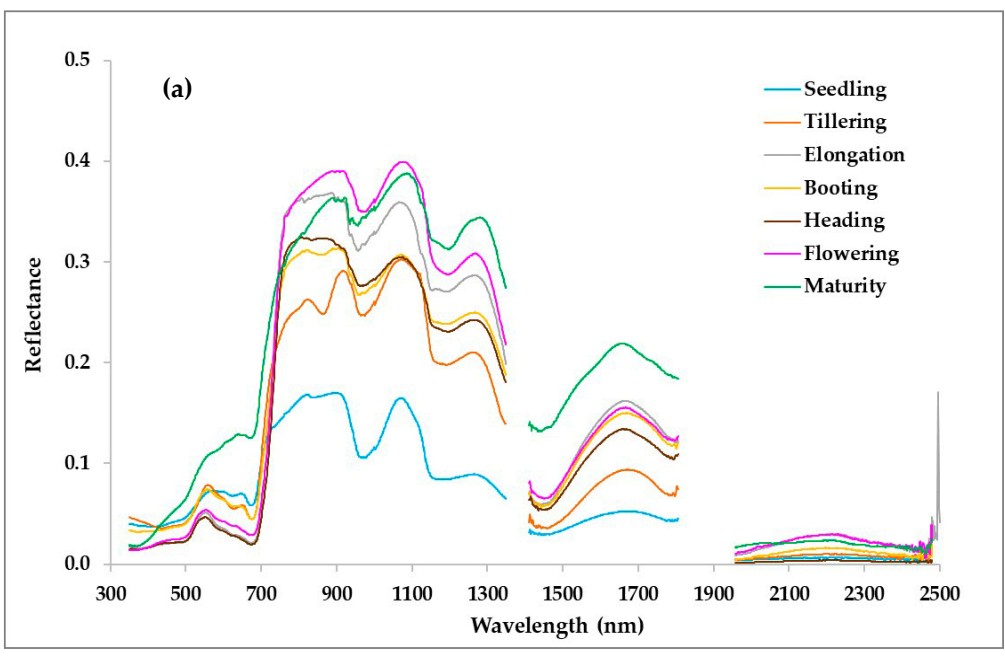

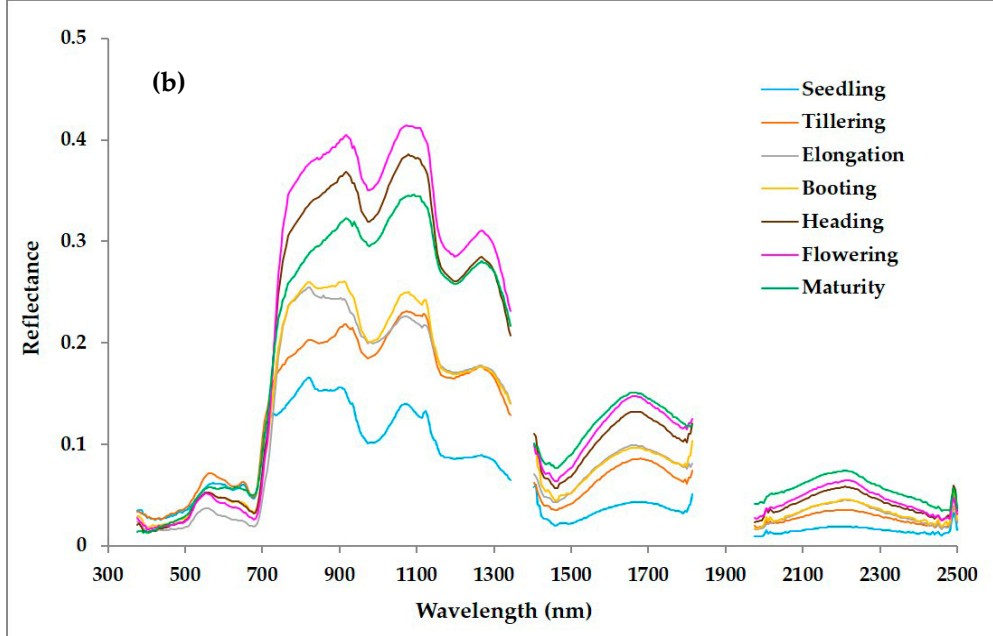

**Figure 4.** Hyperspectral signatures for different crop growth stages of rice recorded using (**a**) ground based measurements and (**b**) AVIRIS-NG.

### 3.2. Reflectance Spectra of Rice Canopy from AVIRIS-NG

The airborne spectral reflectance showed marked variation similar to ground-truth spectra at different phenological stages, as shown in Figure 4b. The variations in spectral reflectance were observed in UV, visible and NIR regions with increased reflectance in NIR region, and reduced reflectance in both UV and visible regions. The findings of canopy reflectance observed in ground-truth spectra were similar and consistent with the airborne spectral reflectance at different phenological stages. In UV region, maximum and minimum reflectance was noticed in tillering and maturity stages, respectively. Within visible region, the reflectance variations were clearly observed in the green region compared to violet blue and red regions, with the highest reflectance in tillering and lowest in elongation stage. In NIR region, the reflectance pattern exhibited increased trend from seedling to booting, and reduced trend from flowering to maturity stage. From these results, it is evident

that various crop growth phases of rice can be differentiated using airborne hyperspectral reflectance spectra. The study on similarity analysis between hyperspectral reflectance of ground-based and airborne showed a SAM score of 0.85 at seedling, 0.84 at tillering, 0.88 at elongation, 0.80 at booting, 0.81 at heading, 0.90 at flowering, and 0.77 at maturity stages of rice (Figure S1).

### 3.3. Rice LAI at Different Phenological Stages

LAI measurements at different phenological stages were presented in Table 3. As the crop advances from seedling to maturity, variations in LAI were obvious with changing phenological stages. LAI increased from seedling to heading, and reached peak at flowering and reduced thereafter (1.21 at seedling to 3.61 at heading and flowering). The range of LAI was maximum at booting (1.70 to 4.00) followed by the tillering (0.75 to 2.60) and elongation stages (1.60 to 3.00). The minimal variations in LAI were observed at the maturity stage (1.60 to 2.50).

**Table 3.** Field measured Leaf Area Index (LAI) of rice at different crop growth stages used for model building.

| Crop Growth Stage | Number of Fields Surveyed | LAI | | | | |
|---|---|---|---|---|---|---|
| | | Mean $\pm$ SD | Minimum | Maximum | *p* Value | CV |
| Seedling | 14 | 1.21 $\pm$ 0.45 | 0.54 | 2.20 | 0.59 | 37.50 |
| Tillering | 19 | 1.71 $\pm$ 0.54 | 0.75 | 2.60 | 0.68 | 31.66 |
| Elongation | 9 | 2.41 $\pm$ 0.53 | 1.60 | 3.00 | 0.12 | 21.79 |
| Booting | 11 | 3.00 $\pm$ 0.82 | 1.70 | 4.00 | 0.04 | 27.24 |
| Heading | 10 | 3.61 $\pm$ 0.47 | 3.10 | 4.30 | 0.15 | 13.00 |
| Flowering | 8 | 3.61 $\pm$ 0.45 | 2.80 | 4.10 | 0.42 | 12.49 |
| Maturity | 7 | 2.11 $\pm$ 0.35 | 1.60 | 2.50 | 0.38 | 16.48 |

### 3.4. Relationship of Vegetation Indices to Rice Phenological Stages

The vegetation indices used in the study showed marked variations with canopy growth from seedling to maturity stage. We used 29 vegetation indices to estimate rice LAI at varied phenological stages (Figure 5, Table A1 in Appendix A). The relationship of different VIs with growth stages from seedling to heading were established. The temporal variations of these VIs increased from seedling to heading, and later decreased till crop maturity. The results in our study revealed that among various indices that were evaluated, NDVI (0.54–0.91), SAVI (0.82–1.36), MSR (0.91–3.54), and VOG (1.06–2.11) found better for different crop growth stages of rice. Similarly, other VIs *viz.*, WDRVI, OSAVI, and GNDVI also performed better from seedling to heading, with the values ranging between 0.50–1.27, 0.38–0.74, and 0.41–0.78, respectively. The SR (3.78–20.79), TVI (7.15–17.95), and RVI (3.83–21.75) showed relatively higher values, whereas MTVI 2 (0.17–0.47), mND$_{705}$ (0.16–0.74), and RDVI (0.27–0.56) exhibited lower values from seedling to heading stage. The vegetation indices SLAIDI, RVSI, MNLI, PRI, DDN, DD, and VOG2 exhibited negative to near zero values across all the rice growth stages.

### 3.5. Evaluation of Vegetation Indices for Rice LAI Estimation

The LAI data collected using hand held canopy analyzer during ground truthing showed higher LAI until the heading stage. Similar results of LAI were observed with AVIRIS-NG data. These vegetation indices were subjected to regression analysis to build models for estimating LAI at each phenological stage. The regression analysis ($R^2$ and RMSE) was performed between LAI and VIs. The indices that are statistically significant at each phenological stage were represented in Figure 6 and Table A2 in Appendix A. The results showed that vegetation indices had significant relationship with LAI until booting stage only. Thereafter, at the heading and flowering stages, the relationship was non-significant. Among the VIs, at seedling stage, SR ($R^2$ = 0.66 **, RMSE = 0.28), MSR ($R^2$ = 0.65 ***, RMSE = 0.28), RVI ($R^2$ = 0.65 ***, RMSE = 0.28), and WDRVI ($R^2$ = 0.64 ***,

RMSE = 0.28) captured LAI better than the other VIs. However, SAVI ($R^2$ = 0.60 **, RMSE = 0.29) and NDVI ($R^2$ = 0.59 **, RMSE = 0.30) exhibited a slightly lower significant relationship with LAI at the same phenological stage. At tillering, though VOG2 ($R^2$ = 0.52 **, RMSE = 0.38) and PRI ($R^2$ = 0.47 **, RMSE = 0.41) had a strong positive relationship, it was not considered as the best fit, since no relationship with LAI was observed at later phenological stages. The $mND_{705}$ exhibited a strong relationship with LAI with better values ($R^2$ = 0.44 **, RMSE = 0.42), followed by NDVI ($R^2$ = 0.42 **, RMSE = 0.42) and SAVI ($R^2$ = 0.42 **, RMSE = 0.42). The DD ($R^2$ = 0.30 **, RMSE = 0.46) and GNDVI ($R^2$ = 0.32 *, RMSE = 0.46) had a weak relationship at all phenological stages.

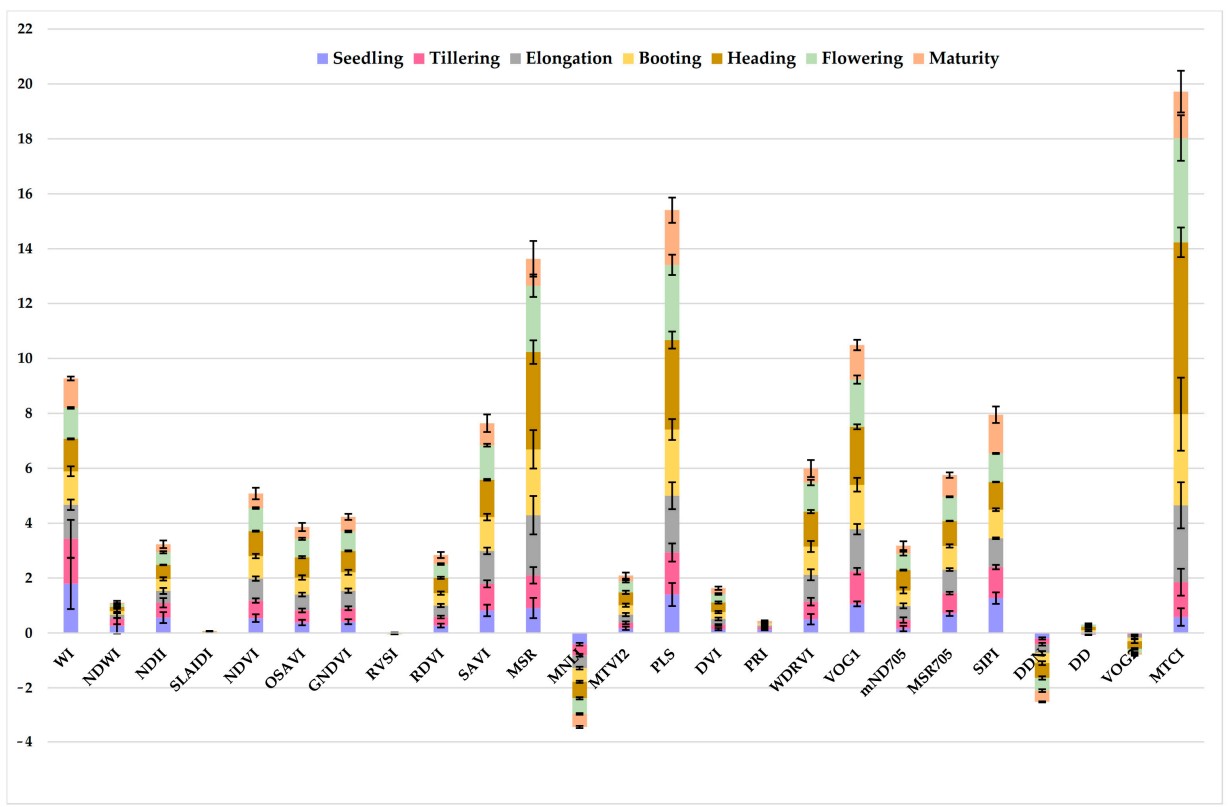

**Figure 5.** Mean of vegetation indices at different crop growth stages of rice estimated from AVIRIS-NG. Error bars represents standard deviation. The mean values in the stacked bar for SR, TVI, RVI, and REP are in the range of 60–750, and thus are not shown in the above Figure (Refer to Table A1 in Appendix A for more details).

Interestingly, REP ($R^2$ = 0.74 **, RMSE = 0.29) and $MSR_{705}$ ($R^2$ = 0.70 **, RMSE = 0.31) had captured LAI better than other VIs during elongation stage, but failed for the rest of the phenological stages. The best-fit models of LAI and VI for elongation stage were NDVI ($R^2$ = 0.69 **, RMSE = 0.31), SAVI ($R^2$ = 0.69 **, RMSE = 0.31), WDRVI ($R^2$ = 0.68 **, RMSE = 0.32), and MSR ($R^2$ = 0.66 **, RMSE = 0.33), while the least fit model was with $mND_{705}$ ($R^2$ = 0.57 *, RMSE = 0.37). At the booting stage, WDRVI ($R^2$ = 0.67 **, RMSE = 0.49), NDVI ($R^2$ = 0.65 **, RMSE = 0.52), SAVI ($R^2$ = 0.65 **, RMSE = 0.51), and MSR ($R^2$ = 0.64 **, RMSE = 0.52) had a strong relationship between LAI and VI, whereas RVI ($R^2$ = 0.53 *, RMSE = 0.59) had a weak relationship at the booting stage. Some of the other Vis *viz.*, RVSI, OSAVI, RDVI, MTVI2, TVI, $mND_{705}$, and PLS were good at seedling stage; SIPI, VOG, MTCI, and DD at tillering; SIPI, PRI, VOG, and MTCI at elongation stage; and SIPI, $MSR_{705}$, OSAVI, RDVI, MTVI2, and DD at booting stage (Figure 6, Table A2 in Appendix A). The indices that had varied responses with different phenological stages was not considered for further validation.

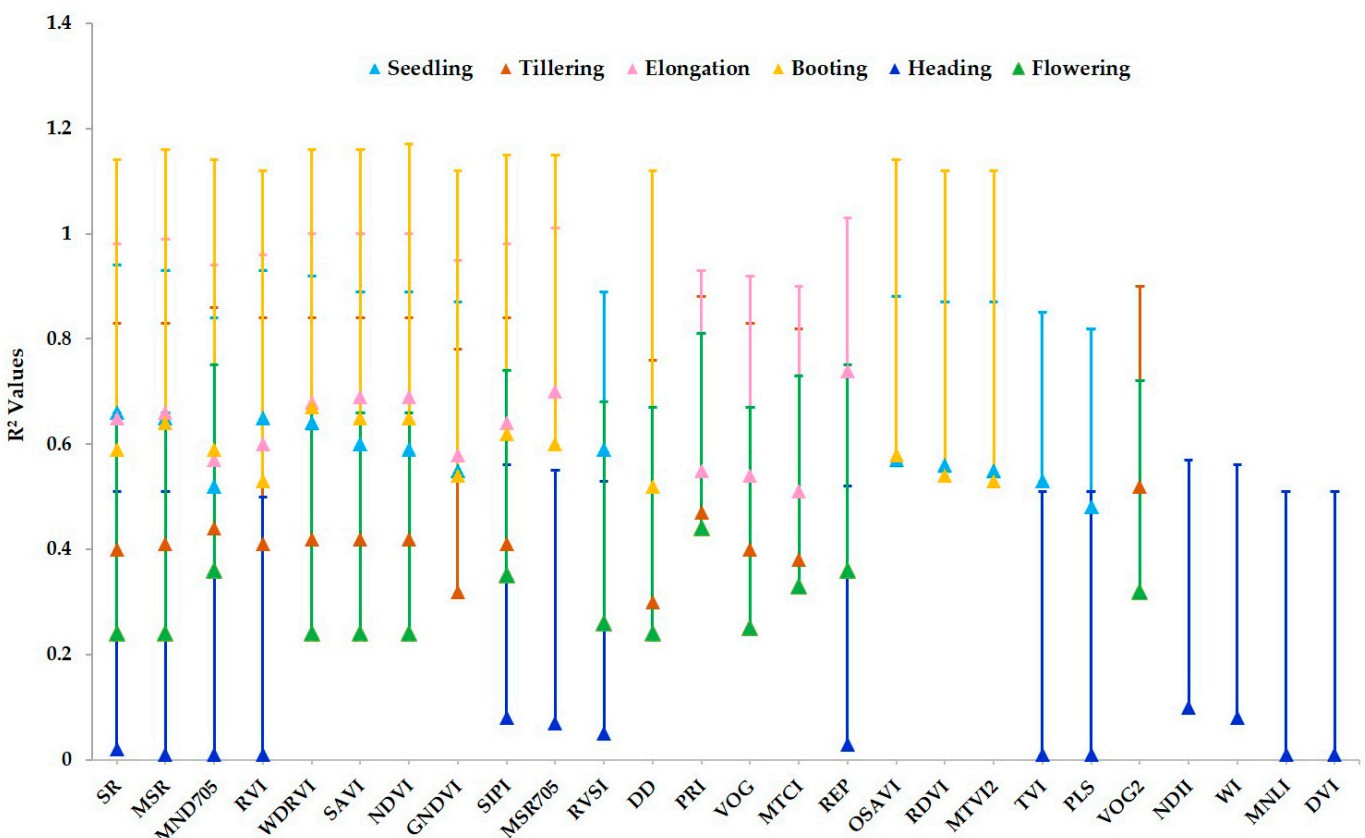

**Figure 6.** Best fit models for estimating LAI using various Vegetation Indices (VI) at different crop growth stages of rice. Error bars represents RMSE values.

### 3.6. Validation of VIs for Estimation of LAI

The best fit VIs were validated using independent data sets of 54 field locations. As the crop growth advanced from seedling to flowering, the LAI increased. Thereafter, it decreased until maturity stage. The maximum deviations in LAI were noticed in the booting stage ranging from 2.50 to 4.10. The minimal variations were observed in the seedling stage, i.e., from 0.60 to 1.31 (Table 4, Figure 7). Further, the $R^2$ and RMSE of the validation dataset at four phenological stages were calculated for all the best-fit models, but only the significant ones are represented in Table 5. Among several VIs that were found statistically significant, only eight VIs were identified to estimate LAI at all the phenological stages except booting stage. For the booting stage, only two VIs were found significant. The $R^2$ and RMSE values differed for each phenological stage and vegetation indices. Similar to the calibration data, VIs showed differential response at each stage with fluctuating $R^2$ values, and the best prediction was observed at the elongation stage.

**Table 4.** Field measured LAI of rice at different crop growth stages used for the validation of models.

| Crop Growth Stage | Number of Fields | LAI | | | | |
|---|---|---|---|---|---|---|
| | | Mean $\pm$ SD | Minimum | Maximum | *p* Value | CV (%) |
| Seedling | 12 | 0.99 $\pm$ 0.23 | 0.60 | 1.31 | 0.33 | 23.51 |
| Tillering | 12 | 1.78 $\pm$ 0.40 | 1.20 | 2.42 | 0.42 | 22.41 |
| Elongation | 7 | 2.89 $\pm$ 0.40 | 2.50 | 3.60 | 0.29 | 13.80 |
| Booting | 7 | 3.29 $\pm$ 0.58 | 2.50 | 4.10 | 0.88 | 17.70 |
| Heading | 6 | 3.63 $\pm$ 0.35 | 3.20 | 4.20 | 0.92 | 9.64 |
| Flowering | 5 | 3.79 $\pm$ 0.57 | 2.90 | 4.50 | 0.42 | 15.11 |
| Maturity | 5 | 2.24 $\pm$ 0.68 | 1.50 | 3.10 | 0.53 | 30.38 |

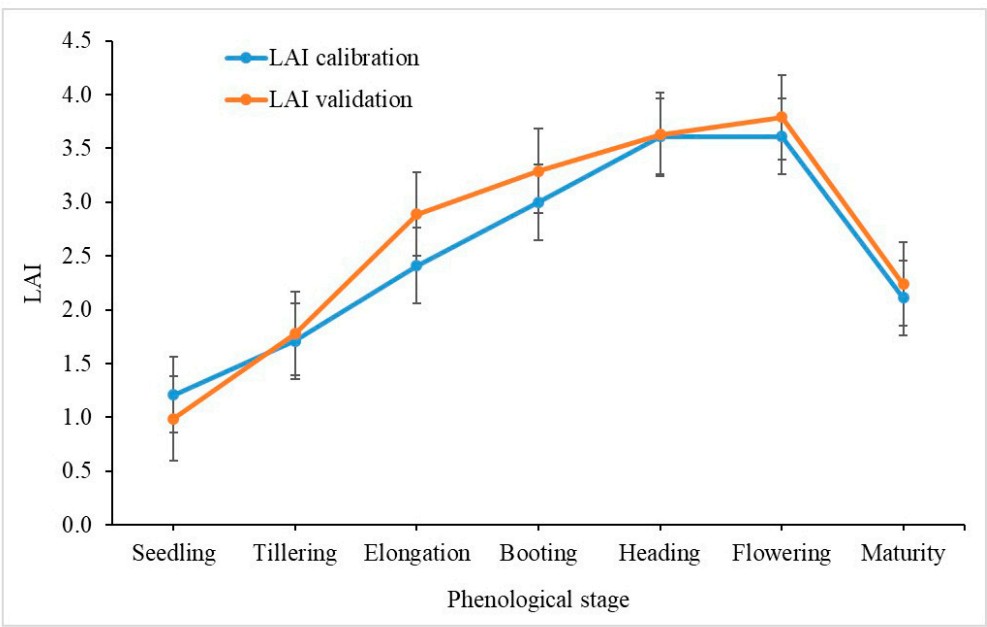

**Figure 7.** LAI of rice fields used for the calibration and validation of data.

**Table 5.** Estimated LAI at different crop growth stages of rice (validation of model outputs).

| VI | Seedling | | Tillering | | Elongation | | Booting | |
|---|---|---|---|---|---|---|---|---|
| | $R^2$ | RMSE | $R^2$ | RMSE | $R^2$ | RMSE | $R^2$ | RMSE |
| $mND_{705}$ | 0.52 ** | 0.17 | 0.42 * | 0.32 | 0.88 ** | 0.15 | 0.51 ns | 0.45 |
| SR | 0.50 ** | 0.17 | 0.40 * | 0.32 | 0.66 * | 0.26 | 0.10 ns | 0.60 |
| MSR | 0.51 ** | 0.17 | 0.48 * | 0.30 | 0.67 * | 0.25 | 0.42 ns | 0.49 |
| RVI | 0.51 ** | 0.17 | 0.38 * | 0.33 | 0.71 * | 0.24 | 0.10 ns | 0.60 |
| WDRVI | 0.47 * | 0.18 | 0.37 * | 0.33 | 0.76 * | 0.22 | 0.62 * | 0.39 |
| SAVI | 0.49 * | 0.17 | 0.56 ** | 0.28 | 0.63 * | 0.27 | 0.59 * | 0.41 |
| NDVI | 0.45 * | 0.18 | 0.34 * | 0.34 | 0.73 * | 0.23 | 0.54 ns | 0.43 |
| GNDVI | 0.49 * | 0.17 | 0.45 * | 0.31 | 0.83 ** | 0.18 | 0.50 ns | 0.45 |

* Significant at 0.05%. ** Significant at 0.01%. ns: non-significant.

At the seedling stage, a strong correlation between LAI and $mND_{705}$ ($R^2 = 0.52$ **) was observed that indicated the best-fit model followed by MSR and RVI ($R^2 = 0.51$ **). Other VIs *viz.*, SR, WDRVI, SAVI, NDVI, and GNDVI, have a significant correlation and have $R^2$ values ranging from 0.45 to 0.50. At the tillering stage, SAVI ($R^2 = 0.56$ **), followed by MSR ($R^2 = 0.48$ *), proved best fit with field LAI. Other Vis which are statistically significant include GNDVI ($R^2 = 0.45$ *), $mND_{705}$ ($R^2 = 0.42$ *), SR ($R^2 = 0.40$ *), WDRVI ($R^2 = 0.37$ *), RVI ($R^2 = 0.38$ *), and NDVI ($R^2 = 0.34$ *). The best estimation of LAI was noticed at the elongation stage ($R^2 = 0.63$–0.88). The best-fit model with strong correlation was noticed between LAI and $mND_{705}$ which had maximum $R^2$ (0.88 **), followed by GNDVI (0.83 **). The performance of Vis, *viz.*, WDRVI, NDVI, and RVI, were better ($R^2 \geqq 0.70$ *) compared to MSR, SR, and SAVI ($R^2 \leq 0.70$ *) at the elongation stage. At the booting stage, only two VIs *viz.*, WDRVI (0.62 *) and SAVI (0.59 *) were found to be strongly correlated with LAI, and the rest of the VIs were non-significant. The satellite images of the best-performing VIs for LAI estimation of the selected fields, *viz.*, $mND_{705}$, SAVI, and WDRVI, were depicted in Figure 8, respectively.

Our results showed that RMSE values also differed with phenological stages. The lower RMSE values were observed at the seedling (0.17–0.18) and elongation stages (0.15–0.27) compared to tillering (0.28–0.34) and booting (0.39–0.60). The models with the least RMSE value will predict better. Thus, the seedling stage can be estimated better followed by the elongation stage. The estimation of LAI with VIs at the booting stage was

not significantly correlated when compared to the other three phenological stages. Hence, most of the indices performed better at later phenological stages (elongation stage, the canopy was fully exposed) compared to early stages (seedling and tillering).

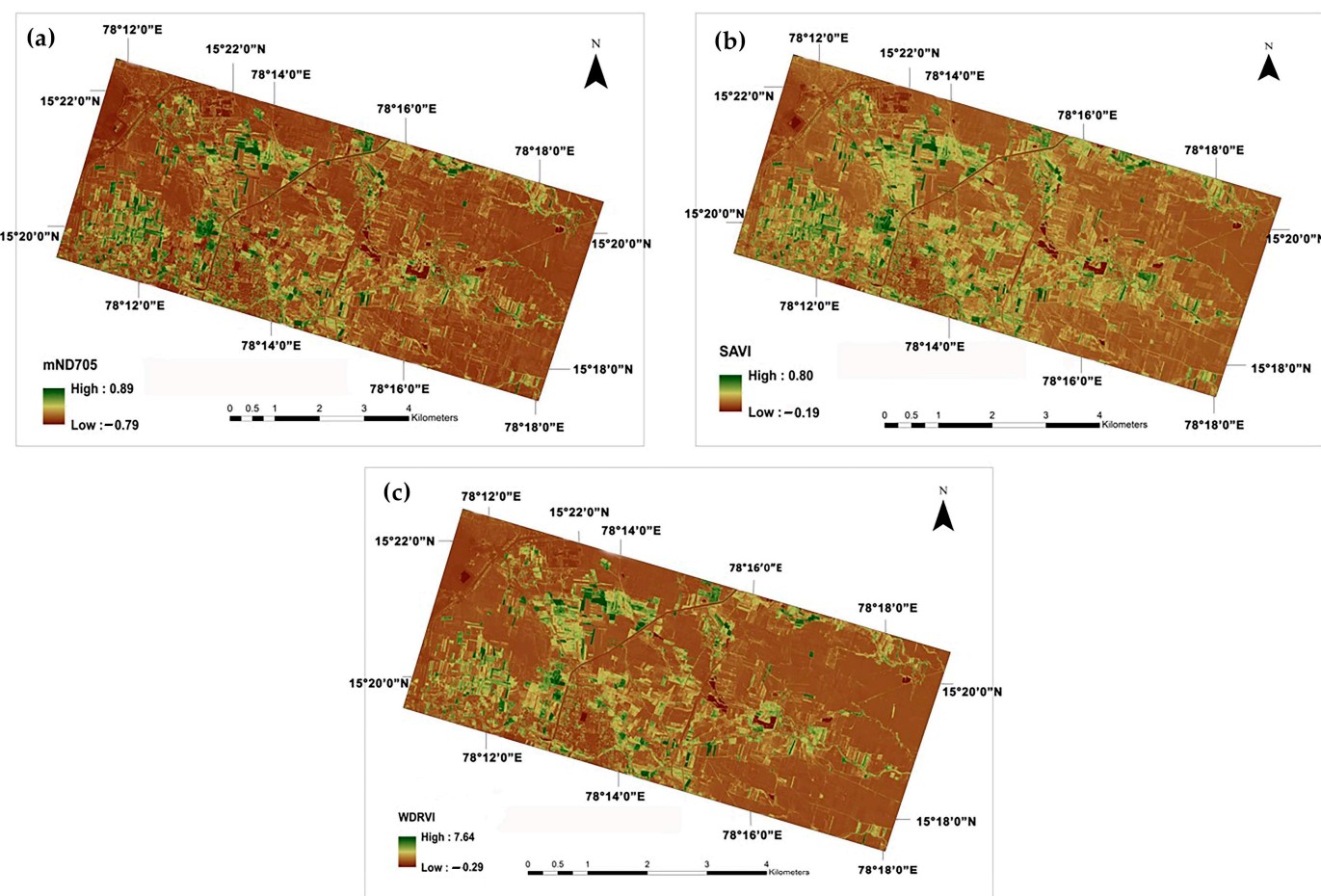

**Figure 8.** Estimated Leaf Area Index (LAI) of rice fields with (**a**) mND$_{705}$; (**b**) SAVI; (**c**) WDRVI.

## 4. Discussion

This study aims to estimate the rice LAI at different phenological stages, *viz.*, seedling, tillering, elongation, booting, heading, flowering, and maturity, using AVIRIS-NG hyperspectral data and ground-measured LAI data. The results indicated that the relationship of LAI with vegetation indices at different phenological stages was distinct (Figure 6, Table A2 in Appendix A). Rice plant undergoes distinct changes in the canopy at different phenological stages during the growing season. As the rice plants grow from seedling to booting, the growth flourishes. Once the plant enters the reproductive stage (heading stage), the panicles start emerging, bringing changes in the canopy architecture. Our findings revealed that LAI increased up to heading, and thereafter decreased (Table 3). The similar changes in LAI at different growth phases of rice has been reported earlier [12,53,54].

The reflectance data of both ground-based hyperspectral radiometry and AVIRIS-NG showed marked variations between phenological stages, and an identical trend was noticed in both the ground-based and airborne hyperspectral data (Figure 4). The results further sowed that the spectral reflectance increased, reached peak value at booting, and reduced thereafter at ripening phase. The canopy reflectance is governed by biophysical characteristics of vegetation, architecture of the canopy, atmospheric absorption, and scattering, direction of incidence radiation, and soil backgrounds [55]. As higher photosynthetic activity takes place during vegetative stage of rice, the plants absorb more photosynthetically active radiation

from the visible region [56], and variations in the reflectance in visible region was low during this stage when compared to heading stage, where the panicles start emerging on the top of canopy (Figure 4b). Contrarily, in the NIR region, the canopy reflectance of rice from ripening stages (heading to maturity) tend to be higher compared to the vegetative stages (Figure 4b). During the early vegetative stage of rice, the canopy reflectance was reduced due to slow growth and increased exposure to soil and water. As the crop advances to reproductive stage, the growth hastens, and increased reflectance was noticed from elongation to flowering, probably due to increased leaf area and reduced exposure of soil [34,57]. Spectral reflectance in the visible region is mainly influenced by pigment absorption, while the reflectance in NIR region is determined by canopy structure, and the changes in leaf orientation from horizontal to vertical at critical stages owes to the overlapping of leaves and chlorotic and necrotic turning during senescence [29,58–61]. The SAM spectral matching score of a avalue close to 1 indicates the best match and higher confidence in the spectral similarity [50]. The SAM score was more than 0.75 at all the growth stages of rice, which indicates the better matching of both ground-based and AVIRIS-NG spectra.

The vegetation indices tested in this study were calculated based on the ratio or difference between NIR and visible reflectance, having a distinct response at different phenological stages. These vegetation indices have been used in the past for estimating biophysical parameters, *viz.*, LAI, chlorophyll, and biomass, in several crops [39,62,63]. Our findings revealed that there was significant correlation between LAI and VIs at different phenological stages (Figure 6, Table A2 in Appendix A), and these results are in accordance with previous studies [64,65]. The ground-measured LAI was best fitted with the VIs at the elongation stage ($R^2$ = 0.51–0.74, $p < 0.05$) and tillering ($R^2$ = 0.30–0.52, $p < 0.05$), followed by the booting stage ($R^2$ = 0.52 to 0.67, $p < 0.05$), suggesting that rice LAI can be predicted better at the elongation stage. This shows that the 51–74% variability in vegetation indices could be explained by LAI, while the remaining 26–49% variation could probably attribute to different agronomic management practices, *viz.*, crop variety, fertilizer management, planting time, etc. Validation of LAI with vegetation indices at different phenological stages showed that the performance was low at seedling ($R^2$ = 0.45–0.52), tillering ($R^2$ = 0.34–0.56), and booting ($R^2$ = 0.10–0.62), whereas better performance was observed at the elongation stage ($R^2$ = 0.63–0.88). The LAI of rice crop at vegetative stages was up to 3.0 and indices, *viz.*, mND$_{705}$, SR, MSR, RVI, WDRVI, SAVI, NDVI, and GNDVI, better predicted LAI across all the stages from seedling to booting, with few exceptions (Table 5), whereas from heading to maturity, there was no relationship between LAI and VIs.

The best-performing VIs for LAI have bands at green, red, blue, and NIR regions. These findings agreed with the earlier reports by Thenkabail et al. [65], Darvishzadeh et al. [66], and Herrmann et al. [67], suggesting that the red-edge and NIR bands are important for LAI assessment and possess strong influence to strengthen the relationship between spectral reflectance and LAI. At seedling and elongation stages, the mND$_{705}$ accurately predicted the LAI, which consists of the combination of red-edge, blue, and NIR bands, suggesting that blue bands can also have influence in LAI estimation [68–70]. This index effectively reduces the impact of differences in leaf surface reflectance and improves the sensitivity of pigment content estimation [71]. At the tillering stage, SAVI with red and NIR bands predicted LAI better, and similar reports were observed by He et al. [72]. This could be because, in contrast to the NDVI, SAVI increases the linearity between the index, and LAI and normalizes the soil-induced changes from spectral vegetation indices without affecting the vegetation measures [21]. At the booting stage, WDRVI with red and NIR bands predicted better LAI with a strong relationship, and explained 62% of the variation in LAI. This might be due to the higher sensitivity of WDRVI to moderate-to-high LAI (between two and six) that aid in monitoring the vegetation status and precision farming [73]. Earlier studies also established the WDRVI vegetation index for accurate estimation of LAI in maize, soybean, wheat, and potato [74,75]. The distinct variation of VIs for LAI at different phenological stages was likely due to the leaf angle within the canopy, rather than individual leaf reflectance properties [63,76].

## 5. Conclusions

In this study, LAI of rice at different growth stages was measured using ground-based data which were compared with the estimations from AVIRIS-NG hyperspectral data. Among 29 hyperspectral vegetation indices evaluated for estimating rice LAI at different growth stages, 8 indices were found promising at seedling, tillering, elongation, and booting stages. Among the stages selected, LAI at the elongation stage was the best predicted using AVIRIS-NG airborne hyperspectral sensors. The findings revealed different vegetation indices for different phenological stages. The SAM spectral matching score was more than 0.75 at all the rice growth stages, indicating the better match between ground-based and AVIRIS-NG spectra. Red-edge, Blue, and NIR bands were found sensitive for seedling and elongation stages, whereas Red and NIR bands were best for the tillering and booting stages. The results revealed that the best suitable hyperspectral vegetation indices for mapping and estimating LAI in rice were $mND_{705}$ at the seedling and elongation stages, and SAVI at tillering and WDRVI at the booting stage. The feasibility of using AVIRIS-NG airborne hyperspectral sensors with very high spatial resolution for the precise mapping of LAI from rice crops is demonstrated in the present study.

**Supplementary Materials:** The following supporting information can be downloaded at: https://www.mdpi.com/article/10.3390/rs16060954/s1, Figure S1: Ground-based and AVIRIS-NG image spectral similarity.

**Author Contributions:** Conceptualization, M.P.; methodology, K.A.G. and U.S.S.; software, G.S.S.; validation, M.T.; formal analysis, M.T., U.S.S. and G.S.K.; investigation, M.P., K.A.G. and N.R.K.; resources, M.P.; data curation, M.T., U.S.S., G.S.K. and P.C.; writing—original draft preparation, U.S.S. and P.C.; writing—review and editing, M.P., K.A.G. and N.R.K.; visualization, M.P.; supervision, M.P.; project administration, V.K.S.; funding acquisition, M.P. All authors have read and agreed to the published version of the manuscript.

**Funding:** This work was funded by Department of Science and Technology, Government of India, Network project Big Data Analytics—Hyperspectral Data (BDA-HSRS), Grant Number BDID/01/23/2014-HSRS18.

**Data Availability Statement:** Data are contained within the article and Supplementary Materials.

**Acknowledgments:** We gratefully acknowledge the farmers for their cooperation during field surveys and collecting ground-truth data. We thankfully acknowledge the support of staff and facilities under the Indian Council of Agricultural Research (ICAR)—National Innovations in Climate Resilient Agriculture (NICRA) project in analyzing the field samples and statistical analysis.

**Conflicts of Interest:** The authors declare no conflicts of interest.

## Appendix A

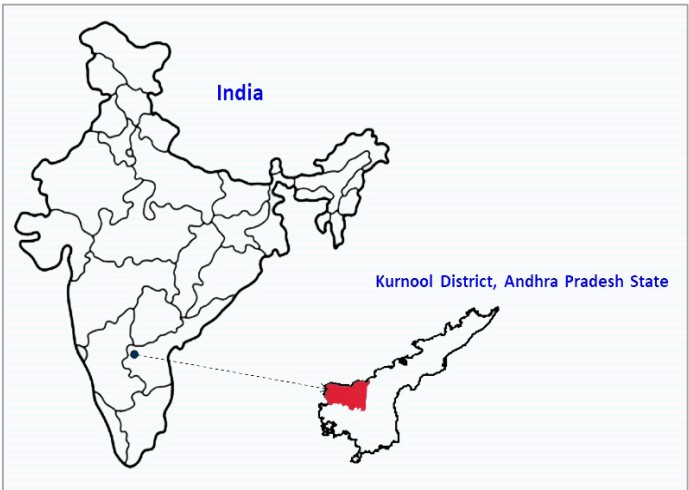

**Figure A1.** Map of India representing the study area.

**Table A1.** Vegetation indices at different crop growth stages of rice estimated from AVIRIS-NG.

| VI | Seedling | Tillering | Elongation | Booting | Heading | Flowering | Maturity |
|---|---|---|---|---|---|---|---|
| WI | $1.80 \pm 0.93$ | $1.63 \pm 0.69$ | $1.24 \pm 0.19$ | $1.22 \pm 0.18$ | $1.18 \pm 0.02$ | $1.13 \pm 0.03$ | $1.07 \pm 0.07$ |
| NDWI | $0.26 \pm 0.28$ | $0.26 \pm 0.20$ | $0.14 \pm 0.11$ | $0.14 \pm 0.08$ | $0.15 \pm 0.01$ | $0.11 \pm 0.03$ | $0.04 \pm 0.07$ |
| NDII | $0.56 \pm 0.20$ | $0.54 \pm 0.17$ | $0.43 \pm 0.11$ | $0.44 \pm 0.06$ | $0.51 \pm 0.01$ | $0.46 \pm 0.04$ | $0.29 \pm 0.14$ |
| SLAIDI | $0.01 \pm 0.00$ | $0.01 \pm 0.00$ | $0.01 \pm 0.00$ | $0.01 \pm 0.00$ | $0.01 \pm 0.00$ | $0.01 \pm 0.00$ | $0.01 \pm 0.00$ |
| NDVI | $0.54 \pm 0.14$ | $0.64 \pm 0.08$ | $0.80 \pm 0.08$ | $0.82 \pm 0.08$ | $0.91 \pm 0.02$ | $0.84 \pm 0.03$ | $0.53 \pm 0.21$ |
| OSAVI | $0.38 \pm 0.10$ | $0.44 \pm 0.07$ | $0.58 \pm 0.07$ | $0.62 \pm 0.08$ | $0.74 \pm 0.04$ | $0.67 \pm 0.04$ | $0.43 \pm 0.15$ |
| GNDVI | $0.41 \pm 0.09$ | $0.49 \pm 0.07$ | $0.64 \pm 0.08$ | $0.67 \pm 0.09$ | $0.78 \pm 0.02$ | $0.71 \pm 0.03$ | $0.53 \pm 0.11$ |
| RVSI | $-0.01 \pm 0.00$ | $-0.01 \pm 0.00$ | $-0.01 \pm 0.00$ | $-0.01 \pm 0.01$ | $0.01 \pm 0.00$ | $-0.00 \pm 0.00$ | $-0.01 \pm 0.00$ |
| REP | $735.60 \pm 64.56$ | $728.59 \pm 8.77$ | $725.35 \pm 1.64$ | $725.43 \pm 2.45$ | $729.34 \pm 0.54$ | $727.22 \pm 1.09$ | $742.50 \pm 16.60$ |
| SR | $3.78 \pm 1.43$ | $4.89 \pm 1.31$ | $10.64 \pm 4.44$ | $12.01 \pm 4.70$ | $20.79 \pm 3.97$ | $11.82 \pm 2.88$ | $4.28 \pm 2.68$ |
| RDVI | $0.27 \pm 0.07$ | $0.31 \pm 0.05$ | $0.42 \pm 0.06$ | $0.45 \pm 0.06$ | $0.56 \pm 0.04$ | $0.50 \pm 0.03$ | $0.33 \pm 0.11$ |
| SAVI | $0.82 \pm 0.21$ | $0.97 \pm 0.13$ | $1.20 \pm 0.12$ | $1.23 \pm 0.12$ | $1.36 \pm 0.03$ | $1.26 \pm 0.05$ | $0.80 \pm 0.32$ |
| MSR | $0.91 \pm 0.37$ | $1.19 \pm 0.30$ | $2.19 \pm 0.70$ | $2.40 \pm 0.70$ | $3.54 \pm 0.43$ | $2.42 \pm 0.41$ | $0.98 \pm 0.65$ |
| TVI | $7.15 \pm 2.86$ | $8.00 \pm 2.18$ | $12.01 \pm 2.47$ | $13.69 \pm 2.33$ | $17.95 \pm 2.07$ | $15.94 \pm 1.54$ | $9.75 \pm 4.18$ |
| MNLI | $-0.41 \pm 0.05$ | $-0.40 \pm 0.05$ | $-0.47 \pm 0.05$ | $-0.50 \pm 0.04$ | $-0.60 \pm 0.04$ | $-0.57 \pm 0.03$ | $-0.51 \pm 0.04$ |
| MTCI | $0.58 \pm 0.32$ | $1.27 \pm 0.49$ | $2.80 \pm 0.84$ | $3.32 \pm 1.33$ | $6.26 \pm 0.54$ | $3.80 \pm 0.83$ | $1.69 \pm 0.76$ |
| MTVI2 | $0.17 \pm 0.06$ | $0.20 \pm 0.05$ | $0.30 \pm 0.06$ | $0.34 \pm 0.06$ | $0.47 \pm 0.06$ | $0.40 \pm 0.04$ | $0.21 \pm 0.11$ |
| PLS | $1.40 \pm 0.42$ | $1.53 \pm 0.33$ | $2.07 \pm 0.49$ | $2.41 \pm 0.38$ | $3.26 \pm 0.31$ | $2.74 \pm 0.37$ | $1.99 \pm 0.46$ |
| RVI | $3.83 \pm 1.49$ | $4.84 \pm 1.22$ | $10.83 \pm 4.54$ | $12.71 \pm 5.72$ | $21.75 \pm 3.91$ | $11.95 \pm 2.90$ | $4.30 \pm 2.73$ |
| DVI | $0.14 \pm 0.04$ | $0.15 \pm 0.03$ | $0.22 \pm 0.05$ | $0.25 \pm 0.04$ | $0.35 \pm 0.04$ | $0.31 \pm 0.03$ | $0.21 \pm 0.06$ |
| PRI | $0.13 \pm 0.03$ | $0.09 \pm 0.02$ | $0.05 \pm 0.03$ | $0.04 \pm 0.03$ | $0.00 \pm 0.01$ | $0.04 \pm 0.02$ | $0.09 \pm 0.02$ |
| WDRVI | $0.50 \pm 0.19$ | $0.64 \pm 0.14$ | $0.98 \pm 0.20$ | $1.03 \pm 0.20$ | $1.27 \pm 0.06$ | $1.06 \pm 0.10$ | $0.51 \pm 0.31$ |
| VOG1 | $1.06 \pm 0.09$ | $1.19 \pm 0.12$ | $1.53 \pm 0.19$ | $1.62 \pm 0.25$ | $2.11 \pm 0.09$ | $1.72 \pm 0.15$ | $1.26 \pm 0.19$ |
| $mND_{705}$ | $0.16 \pm 0.10$ | $0.31 \pm 0.10$ | $0.52 \pm 0.09$ | $0.56 \pm 0.11$ | $0.74 \pm 0.02$ | $0.60 \pm 0.07$ | $0.29 \pm 0.16$ |
| $MSR_{705}$ | $0.71 \pm 0.09$ | $0.74 \pm 0.05$ | $0.86 \pm 0.05$ | $0.86 \pm 0.06$ | $0.91 \pm 0.01$ | $0.88 \pm 0.02$ | $0.79 \pm 0.10$ |
| SIPI | $1.27 \pm 0.21$ | $1.13 \pm 0.08$ | $1.05 \pm 0.03$ | $1.04 \pm 0.05$ | $1.01 \pm 0.01$ | $1.04 \pm 0.02$ | $1.41 \pm 0.30$ |
| DDN | $-0.20 \pm 0.04$ | $-0.21 \pm 0.05$ | $-0.32 \pm 0.07$ | $-0.37 \pm 0.07$ | $-0.54 \pm 0.06$ | $-0.46 \pm 0.05$ | $-0.41 \pm 0.02$ |
| DD | $-0.05 \pm 0.01$ | $-0.01 \pm 0.02$ | $0.04 \pm 0.03$ | $0.06 \pm 0.04$ | $0.13 \pm 0.02$ | $0.08 \pm 0.02$ | $0.00 \pm 0.04$ |
| VOG2 | $-0.02 \pm 0.01$ | $-0.04 \pm 0.02$ | $-0.11 \pm 0.04$ | $-0.13 \pm 0.07$ | $-0.28 \pm 0.03$ | $-0.16 \pm 0.04$ | $-0.05 \pm 0.04$ |

The values are Mean ± Standard deviation.

**Table A2.** Best fit models for estimating LAI using different Vegetation Indices (VI) at different crop growth stages of rice.

| VI | Seedling | | VI | Tillering | | VI | Elongation | | VI | Booting | | VI | Heading | | VI | Flowering | |
|---|---|---|---|---|---|---|---|---|---|---|---|---|---|---|---|---|---|
| | $R^2$ | RMSE | | $R^2$ | RMSE | | $R^2$ | RMSE | | $R^2$ | RMSE | | $R^2$ | RMSE | | $R^2$ | RMSE |
| SR | 0.66 *** | 0.28 | VOG2 | 0.52 *** | 0.38 | REP | 0.74 ** | 0.29 | WDRVI | 0.67 ** | 0.49 | NDII | 0.10 ns | 0.47 | PRI | 0.44 ns | 0.37 |
| MSR | 0.65 *** | 0.28 | PRI | 0.47 ** | 0.41 | MSR$_{705}$ | 0.70 ** | 0.31 | NDVI | 0.65 ** | 0.52 | WI | 0.08 ns | 0.48 | REP | 0.36 ns | 0.39 |
| RVI | 0.65 *** | 0.28 | mND$_{705}$ | 0.44 ** | 0.42 | NDVI | 0.69 ** | 0.31 | SAVI | 0.65 ** | 0.51 | SIPI | 0.08 ns | 0.48 | mND$_{705}$ | 0.36 ns | 0.39 |
| WDRVI | 0.64 *** | 0.28 | NDVI | 0.42 ** | 0.42 | SAVI | 0.69 ** | 0.31 | MSR | 0.64 ** | 0.52 | MSR$_{705}$ | 0.07 ns | 0.48 | SIPI | 0.35 ns | 0.39 |
| SAVI | 0.60 ** | 0.29 | SAVI | 0.42 ** | 0.42 | WDRVI | 0.68 ** | 0.32 | SIPI | 0.62 ** | 0.53 | RVSI | 0.05 ns | 0.48 | MTCI | 0.33 ns | 0.40 |
| NDVI | 0.59 ** | 0.30 | WDRVI | 0.42 ** | 0.42 | MSR | 0.66 ** | 0.33 | MSR$_{705}$ | 0.60 ** | 0.55 | REP | 0.03 ns | 0.49 | VOG2 | 0.32 ns | 0.40 |
| RVSI | 0.59 ** | 0.30 | MSR | 0.41 ** | 0.42 | SR | 0.65 ** | 0.33 | SR | 0.59 ** | 0.55 | SR | 0.02 ns | 0.49 | RVSI | 0.26 ns | 0.42 |
| OSAVI | 0.57 ** | 0.31 | RVI | 0.41 ** | 0.43 | SIPI | 0.64 * | 0.34 | mND$_{705}$ | 0.59 ** | 0.55 | MSR | 0.01 ns | 0.50 | VOG | 0.25 ns | 0.42 |
| RDVI | 0.56 ** | 0.31 | SIPI | 0.41 ** | 0.43 | RVI | 0.60 * | 0.36 | OSAVI | 0.58 ** | 0.56 | TVI | 0.01 ns | 0.50 | NDVI | 0.24 ns | 0.42 |
| GNDVI | 0.55 ** | 0.32 | SR | 0.40 ** | 0.43 | GNDVI | 0.58 * | 0.37 | GNDVI | 0.54 * | 0.58 | MNLI | 0.01 ns | 0.50 | SR | 0.24 ns | 0.42 |
| MTVI2 | 0.55 ** | 0.32 | VOG | 0.40 ** | 0.43 | mND$_{705}$ | 0.57 * | 0.37 | RDVI | 0.54 * | 0.58 | PLS | 0.01 ns | 0.50 | SAVI | 0.24 ns | 0.42 |
| TVI | 0.53 ** | 0.32 | MTCI | 0.38 ** | 0.44 | PRI | 0.55 * | 0.38 | MTVI2 | 0.53 * | 0.59 | RVI | 0.01 ns | 0.49 | MSR | 0.24 ns | 0.42 |
| mND$_{705}$ | 0.52 ** | 0.32 | GNDVI | 0.32 * | 0.46 | VOG | 0.54 * | 0.38 | RVI | 0.53 * | 0.59 | DVI | 0.01 ns | 0.50 | WDRVI | 0.24 ns | 0.42 |
| PLS | 0.48 ** | 0.34 | DD | 0.30 * | 0.46 | MTCI | 0.51 * | 0.39 | DD | 0.52 * | 0.60 | mND$_{705}$ | 0.01 ns | 0.50 | DD | 0.24 ns | 0.43 |

* Significant at 0.05%. ** Significant at 0.01%. *** Significant at 0.001%. ns: non-significant.

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
