# Peer review of "Mapping Leaf Area Index at Various Rice Growth Stages in Southern India Using Airborne Hyperspectral Remote Sensing"

_remotesensing, doi:10.3390/rs16060954_

Round 1

Reviewer 1 Report

Comments and Suggestions for Authors

The study effectively used hyperspectral remote sensing, including airborne AVIRIS-NG and ground-based FieldSpec 3 Hi-Res spectroradiometer, to monitor rice Leaf Area Index (LAI) in Southern India, identifying specific vegetation indices that accurately estimate LAI at different growth stages. The study is generally acceptable, but in my opinion needs a few minor corrections. The authors are expected to answer the following questions:

1. What do "viz." and "mt" mean in the introduction? 

2. The introduction should also mention similar studies in the literature on rice cultivation using remote sensing.

3. The resolution of Figure 1 should be increased and FCC should be mentioned in the figure legend.

4. Full names as well as abbreviations of vegetation names should be written.

5. Add a reference to the sentence "The spectral reflectance data from 1300-1400 nm ..." starting on page line 133.

6. Which regression analysis was used and was only one regression analysis used?

7. The resolution of all figures is quite low, all of them need to be improved.

8. Starting on page line 211 "Results showed that NDVI (0.54-0.91), SAVI (0.82-1.36), MSR (0.91- 3.54) and VOG (1.06-2.11) performed better." What does the beginning sentence mean. In addition, table 4 should be interpreted in a more data-driven way and the relationship between growing seasons and indices should be interpreted.

9. What does "which was also found true" mean in the sentence starting on page line 221 "The LAI data collected using hand held canopy analyzer during ground truthing..."?

10. What is the method of atmospheric correction of AVIRIS-NG hyperspectral airborne data?

11. The similarity between Figures 4a and 4b should be compared with a statistical method such as correlation.

12. In the Conclusion section, the results of the study should be mentioned in more detail.

Comments on the Quality of English Language

Minor editing of English language required

Author Response

Thank you for your time and efforts in providing the comments in the manuscript. We appreciate your critical comments that helped us in improving the manuscript. We tried to incorporate all the suggestions by you. 

Reviewer 2 Report

Comments and Suggestions for Authors

In general the paper is well written and communicated. Nevertheless some major comments are needed. 

References are old dated: some new reference would be needed especially in the literature review.

Ground truth data have been collected in different dates and at different times. How did you calibrate data, and how did you compensate possible bias arising at different time of the day and in different days?

Airborne Data have been taken at a very high altitude: how did you corrected for distortions due to air humidity, and air density?

Authors make use on a quite large amount of data: please add the "digitization footprint" of the experiment or at least of the airborn data (maybe as Mb/ha). 

Standard deviation, and thus also measurement uncertainty is quite high and probably too high to allow distinguishing different phenological or management phases. Thus the authors should better explain how the reported results might be in practice used at local farm or at district level. 

Data refer to 2018: have you collected other data in the meanwhile, to confirm or extend the value of results?

Other minor comments:

In Table 2. there is no need to report one reference for each hyperspectral vegetation index: I would refer to just to a couple of reviews summarising all (or the majority) of indices). 

I would move Table 4 and 5 to the appendix, and I would put two graphs in their place in the main text. 

Figure 1 is unreadable: please increase the quality/resolution. 

Figure 2 is not needed: please remove. 

Figure 6: increase font size: as it is, it is unreadable

Comments on the Quality of English Language

English is fine

Author Response

We thank you for the efforts in critically reviewing the manuscript and providing valuable comments. We tried to address all the comments which helped to improve the manuscript.

Round 2

Reviewer 1 Report

Comments and Suggestions for Authors

Necessary corrections were made by the authors. It is suitable for publication.

Author Response

Thank you for accepting our corrections and recommending it for publication. We once again greatly appreciate your time and efforts in reviewing the manuscript.

Reviewer 2 Report

Comments and Suggestions for Authors

The paper has been satisfactorily improved, and I believe the paper is now ready for publication. 

Comments on the Quality of English Language

English is fine

Author Response

Thank you for providing critical inputs that helped us in improving the manuscript and recommending the revised manuscript for publication.